

# Improved machine-learning based open-water/sea-ice/cloud discrimination over wintertime Antarctic sea ice using MODIS thermal-infrared imagery

Stephan Paul[1] and Marcus Huntemann[2,3]

[1]Department of Geography, Ludwig-Maximilian's-University Munich, Munich, Germany
[2]Department of Environmental Physics, University of Bremen, Bremen, Germany
[3]Alfred Wegener Institute, Helmholtz Centre for Polar and Marine Research, Bremerhaven, Germany

**Correspondence:** Stephan Paul (stephan.paul@lmu.de)

**Abstract.** The frequent presence of cloud cover in polar regions limits the use of the Moderate-Resolution Imageing Spectro-radiometer (MODIS) and similar instruments for the investigation and monitoring of sea-ice polynyas compared to passive-microwave-based sensors. The very low thermal contrast between present clouds and the sea-ice surface in combination with the lack of available visible and near-infrared channels during polar nighttime results in deficiencies in the MODIS cloud mask

and dependent MODIS data products. This leads to frequent misclassifications of i) present clouds as sea ice and ii) open-water/thin-ice areas as clouds, which results in an underestimation of polynya area and subsequently derived information. Here, we present a novel machine-learning based approach using a deep neural network that is able to reliably discriminate between clouds, sea-ice, and open-water/thin-ice areas in a given swath solely from thermal-infrared MODIS channels and additionally derived information. Compared to the reference MODIS sea-ice product, our data results in an overall increase

of 31 % in annual swath-based coverage, attributed to an improved cloud-cover discrimination. Overall, higher spatial coverage results in a better sub-daily representation of thin-ice conditions that cannot be reconstructed with current state-of-the-art cloud-cover compensation methods.

## 1  Introduction

Information on cloud presence is of crucial importance when using thermal-infrared imagery. This is especially true for the polar regions, where the thermal contrast between clouds and the underlying snow and sea-ice surface can be low due to persistent surface temperature inversion and low clouds (Welch et al., 1992). Furthermore, occurrences of warm clouds over cold sea ice as well as cold clouds over relatively warm and thin sea ice are both possible. Despite improvements (Liu et al., 2004; Frey et al., 2008; Holz et al., 2008; Liu and Key, 2014), the performance of the frequently used Moderate-Resolution

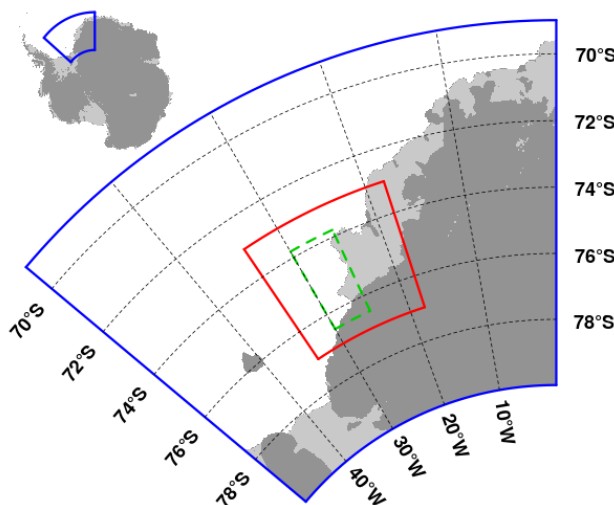

**Figure 1.** Location of the general (red) and focus (green) study area of the Antarctic Brunt Ice Shelf in the south-eastern Weddell Sea (blue). Data of land ice (dark gray) and floating ice shelves (light gray) are retrieved from Rtopo-2 (Schaffer et al., 2016).

Imaging Spectroradiometer (MODIS) cloud mask product (MOD35/MYD35; Ackerman et al., 2015) is substantially reduced during polar nighttime compared to its performance during daytime conditions.

Nonetheless, several studies use MODIS thermal-infrared (TIR) data to monitor polyanya area and associated sea-ice production in polynyas both, in the Arctic as well as the Antarctic and are comparable to or even outperform studies using passive-microwave satellite data in certain regions (e.g., Paul et al., 2015; Preußer et al., 2019). These studies generally utilize

the ice-surface temperature from the National Snow and Ice Data Center (NSIDC) sea-ice product (MOD/MYD29; Hall et al., 2004; Hall and Riggs, 2015a, b). The MOD/MYD29 product is derived from both MODIS sensors onboard the NASA polar orbiting Aqua and Terra satellites with the MOD/MYD35 cloud mask product already applied (Riggs and Hall, 2015). However, especially positive temperature-anomaly features such as large warm open-water areas through sea-ice polynyas pose a problem for the MODIS cloud mask and result in frequent misclassification of these areas as cloud cover.

In this study, we propose a new machine-learning based approach to discriminate between open-water/thin-ice, sea-ice and cloud-covered areas in MODIS TIR swaths. We evaluate and analyze the use of two different approaches – Random Forests (Breiman, 2001) and a Neural Network (e.g., Kohonen, 1988) – building upon a comprehensive set of newly generated labeled training data. The data set is derived using a combined approach of unsupervised deep-learning, subsequent clustering, and manual screening from co-located full channel 1km resolution MOD/MYD02 product data (MODIS Characterization

Support Team (MCST), 2017a, b) accessed through the Level-1 and Atmosphere Archive & Distribution System (LAADS) Distributed Active Archive Center (DAAC) and Sentinel-1 A/B (S1-A/B) synthetic aperture radar (SAR) calibrated backscatter data acessed through the Alaska Satellite Facility (ASF) DAAC as a cloud-independent reference.



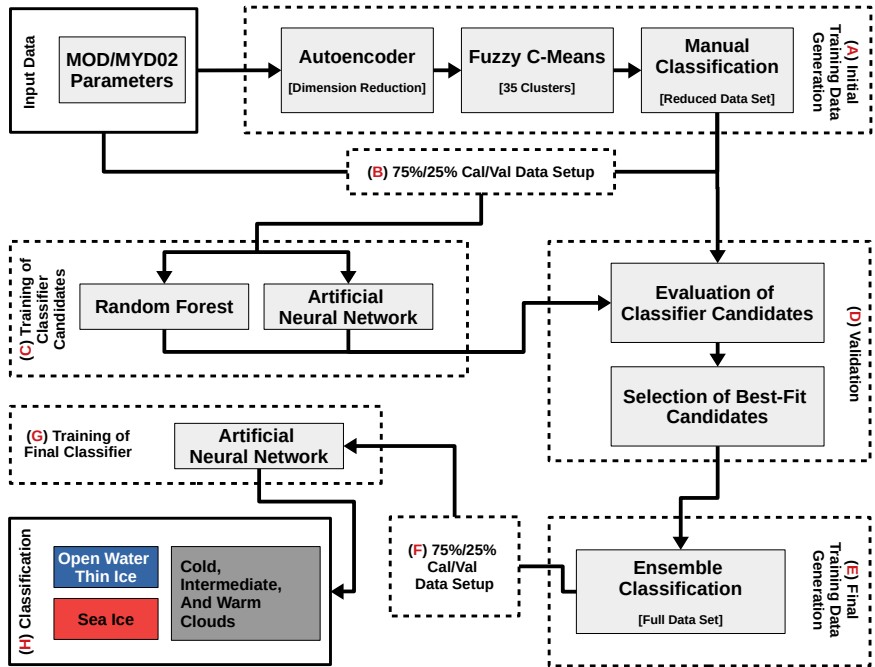

**Figure 2.** Flow chart summarizing all processing steps from the generation of the initial training data through manual classification to the training of the final classifier and its application for open-water/sea-ice/cloud discrimination.

The resulting classifier performance is then analyzed and evaluated based on wintertime estimates of resulting polynya area in comparison to the MOD/MYD29 reference product for the Brunt Ice Shelf (BIS) region in the Antarctic Weddell Sea in the year 2017 (Fig. 1).

In the following sections, we will first describe our methodology and input data starting with the employed basic methods and algorithms (Sect. 2.1) as well as the used input data (Sect. 2.2), a detailed explanation of the initial training data generation scheme (Sect. 2.3) and the subsequent training and evaluation of the initial supervised classifiers (Sect. 2.4&2.5). Next, we describe the steps that lead to our final classifier (Sect. 2.6) and, finally, describe and discuss our results (Sect. 3) in comparison to standard MOD/MYD29-derived estimates as well as using co-located S1-A/B SAR reference data. In the end we provide a summary and an outlook to future applications (Sect. 4).

## 2 Data and Methods

In the following subsections we describe our methods and input data that lead to our deep neural network for the sea-ice, cloud, and open-water/thin-ice discrimination (Fig. 2).



## 2.1 Basic methods and algorithms

This section intends to provide a basic introduction to the methods used in this study. However, it would be beyond the scope of this article to provide an exhaustive review of these methods. Please refer to the provided references for more details.

All computations for this study were carried out using the R software (R Core Team, 2018) running on a commercially available laptop.

### 2.1.1 Gray-level co-occurrence matrices (GLCM)

Gray-level co-occurrence matrices (GLCM) are a tool to quantify spatial texture based on brightness values of a pixel neighborhood (Haralick et al., 1973; Haralick, 1979; Hall-Beyer, 2017; R: Zvoleff, 2019). The directional-dependent occurrence frequencies of brightness-value combinations are counted and normalized to probabilities. Subsequently, several statistical measures can be calculated from the GLCM as an additional descriptive statistic of the data.

Haralick et al. (1973) proposed 14 different metrics. However, not all were commonly adopted and implemented into modern software. For R, eight different measures are implemented (Zvoleff, 2019), from which we utilized three: GLCM Mean, GLCM Variance, and Homogeneity (Tab. 1). Hall-Beyer (2017) showed that GLCM variance can be associated with edges of different class patches, while GLCM mean and Homogeneity correspond well to patch-interior texture.

In general, the use of GLCM texture metrics is suitable for cloud detection and classification in polar regions using visual, near-, and thermal infrared satellite data (Welch et al., 1992). However, as the size of each GLCM per pixel in a sliding-neighborhood window corresponds and increases proportionally to the image bit depth, computational cost increases rapidly for i) large sliding windows and ii) large number of gray-levels in the input data. For our study, all MOD/MYD02 channel-based input parameters for the GLCM computations were re-scaled to 32 gray-levels, using a $3 \times 3$ sliding-neighborhood window with horizontal, vertical as well as diagonal directional pixel relationships.

### 2.1.2 Fuzzy c-means clustering (FCM)

For initial clustering of our data, we utilize an unsupervised procedure called fuzzy c-means clustering (FCM; Dunn, 1973; Bezdek et al., 1984; R: Meyer et al., 2019).

The FCM is comparable to a classic k-means clustering approach (MacQueen, 1967; Hartigan and Wong, 1979), with the addition of providing cluster membership probabilities for each pixel. In contrast to 'hard'-clustering approaches such as k-means, FCM allows a pixel to belong to several clusters with a certain probability. This type of clustering is also referred to as 'soft' clustering.

For this type of unsupervised partitioning clustering, it is necessary to preselect the number of clusters in which the input data should be separated into. Without a-priori knowledge about potential relationships and correlations between predictors, it is standard practice to choose a large number of initial clusters and manually merge similar clusters afterwards to the desired number of classes.

In this study, we always use a setup of 35 clusters and stop the clustering process after 30 iterations.



### 2.1.3 Artificial neural networks (NN)

An artificial neural network (NN) generally consists of several neurons organized in hierarchical layers in which each neuron of a layer is fully interconnected to all neurons in the adjacent two layers through weighted paths. These neurons respond
to the weighted input of the preceding neurons and pass on their output to the adjacent neurons, modulated based on a type of activation function (Kohonen, 1988; Lee et al., 1990; Welch et al., 1992; Atkinson and Tatnall, 1997; LeCun et al., 2015; Schmidhuber, 2015; R: Lammers, 2019).

Once trained, NN are powerful tools for fast and efficient processing of large amounts of remote sensing data and have been shown to be more accurate, e.g., in classification tasks, than other techniques (Kohonen, 1988; Lee et al., 1990; Atkinson and
Tatnall, 1997).

Furthermore, NN can represent complex and non-linear functions without formal description through learning from labeled training data. In contrast to statistical methods, NN allow to incorporate data from different sources and require no knowledge or assumptions about its parametric distributions. Hence, in contrast to statistical approaches, NN solely depend on provided input data (Lee et al., 1990; Atkinson and Tatnall, 1997; LeCun et al., 2015).

In their simplest form, a so-called 'shallow' NN consists of an input layer, a hidden layer and an output layer. Input-layer neurons correspond to the number of input features/predictors, whereas output layer neurons correspond in our case to the number of classes the input data should be categorized into. With an increasing number of hidden layers, so-called 'deep' NN can handle even more complex problems (Atkinson and Tatnall, 1997; Schmidhuber, 2015).

In this study, we experiment with different numbers of hidden layers, activation functions, and the number of neurons
per hidden layer. While some general suggestions for the NN architecture exist, solutions are often found empirically by maximizing the accuracy through validation data classification. This process is described in the following subsections. For all NN in this study, we use a logarithmic loss function and the adam optimizer (Kingma and Ba, 2014) as well as either rectified linear unit (ReLU) or hyberbolic tangent (TanH) activation functions (LeCun et al., 2015).

In addition to these general NN, we work with a second type called an autoencoder (AE). An AE is a specialized variant of
a NN used for anomaly detection and dimension reduction (Cao et al., 2018; Dong et al., 2018; R: Lammers, 2019).

In a typical AE, the output or target data is equal to the input data. However, all information is forced through a bottle-neck hidden layer. The result relies on the capability of the bottle-neck hidden-layer neurons to extract relevant features from the training data to enable the AE to reconstruct the input image with minimized error (Cao et al., 2018).

This is achieved by constructing two branches of symmetric hidden layers of neurons (called the encoder and the decoder,
respectively) around a bottle-neck neuron layer generally consisting of very few neurons (Cao et al., 2018). The resulting encoder part of the AE can then be used for dimension reduction.

In this study, we trained the AE in a **39-23-10-3**-10-23-39 fully-coupled hidden-layer setup with ReLU activation functions in 35 epochs, using a batch size of 100 and a pseudo-huber loss function (Charbonnier et al., 1997). The used parameters for the AE training and its application as well as for the NN are summarized in Table 1.





### 2.1.4 Random forests (RF)

Random Forests (RF) are an ensemble supervised machine-learning method based on multiple decision trees (Breiman, 2001; R: Liaw and Wiener, 2002). Each decision tree in a RF is a rather simple statistical tool to predict data categories based on thresholds used in several steps to split the input data. When visualized, a decision tree resembles a tree with an increasing number of branches between nodes, leading to final categories.

During the RF training, all single decision trees are fitted to the input training data following three rules/constraints (Breiman, 2001): i) For each decision tree in a RF, training data of the same size as the input data is randomly sampled with replacement from the input data. ii) For $M$ input parameters ideally a fixed number $m \ll M$ is specified and randomly selected out of $M$. The best split on these selected parameters $m$ is then used to split the training data at a specific node. Throughout the growth of the RF, the value of $m$ is held constant. iii) Each tree is grown out fully with no pruning applied (i.e., data is split until each data point is categorized).

In contrast to single decision trees that tend to overfit (i.e., match data too precisely and therefore fail for any additional data), RF do not overfit and are also capable of dealing with unbalanced data sets (Breiman, 2001).

An additional benefit of the RF is its internally calculated importance measure for each provided predictor. In other words, it counts the frequency of use of each predictor in the growth process of each tree as well as the loss in overall accuracy when dropping this predictor. Furthermore, similar to the NN approach, RF are also capable to provide probabilistic classification in addition to a standard binary classification (Liaw and Wiener, 2002).

### 2.2 Input data

In total, we use four different types of data sets for the year 2017:

1. MODIS Level 1B Calibrated Radiances obtained from the MODIS sensors on-board the polar-orbiting NASA satellites Terra and Aqua (MOD/MYD02; MODIS Characterization Support Team (MCST), 2017a, b; retrieved from the LAADS DAAC at: https://ladsweb.modaps.eosdis.nasa.gov/) with a spatial resolution of 1 km × 1 km at nadir and swath dimensions of 1354 km (across track) × 2030 km (along track),

2. Sentinel-1 A/B Level 1 calibrated backscatter data (S1-A/B; retrieved from the ASF DAAC at: https://https://asf.alaska.edu/ and processed by ESA) with a spatial resolution of 20 m × 20 m,

3. NSIDC MODIS Sea Ice product (MOD/MYD29, Hall et al., 2004; Riggs and Hall, 2015) comprising a pre-computed and MODIS cloud-masked applied ice-surface temperature data set (IST), as well as

4. ECMWF ERA-Interim atmospheric reanalysis data (Dee et al., 2011) featuring a spatial resolution of 0.75 ° and a temporal resolution of 6 h.

An overview of all used input parameters with their respective source as well as their application is provided in Table 1.

All MODIS and ERA-Interim data are resampled to a common equi-rectangular grid of the Brunt Ice Shelf (BIS) area with an average spatial resolution of 1 km × 1 km and an extent from 34 °W to 18 °W and 77 °S to 73 °S using a nearest-neighbor





**Table 1.** Summary of all used parameters, their source product/sensor as well as their application in this study. These parameters comprise brightness temperatures ($BT$) from the selected MOD/MYD02 channel subset ($\star$), as well as BT differences ($\Delta BT$) to the neighboring swaths ($\bullet$; i.e. previous or next swath covering the same area) as well as normalized BT ($BT_{norm}$; $\diamond$). Furthermore, ice-surface temperatures (IST) from MOD/MYD02 together with the IST from neighboring swaths ($IST_{Neighbors}$) as well as IST from the MOD/MYD29 product. The texture metrics calculated from GLCM (Mean, Variance, and Homogeneity), as well as the calibrated backscatter ($\sigma^0$) from Sentinel-1 A/B as reference (R). Finally, the atmospheric parameters taken from the ERA-Interim reanalysis necessary for the calculation of thin-ice thickness (TIT). The applications comprise primarily their use in Neural Network (NN) or Random Forest (RF) classifier and autoencoder (AE) training. Bold **NN** marks the parameters used for the final classifier.

| Symbol/Abbreviation | Parameter | Source | Application |
| --- | --- | --- | --- |
| $BT$ [$\star$] | Brightness Temperatures | MOD/MYD02 | **NN**/RF |
| $\Delta BT$ [$\star$/$\bullet$] | Brightness Temperature Differences | MOD/MYD02 | AE |
| $BT_{norm}$ [$\star$/$\diamond$] | Normalized Brightness Temperatures | MOD/MYD02 | AE/**NN**/RF |
| $IST$ | Ice-Surface Temperature | MOD/MYD02 | AE/**NN**/RF + TIT |
| $IST_{Neighbors}$ | Ice-Surface Temperature of neighbor swaths | MOD/MYD02 | AE/**NN**/RF |
| $GLCM_{Mean}$ [$\star$] | Mean of the GLCM | MOD/MYD02 | **NN**/RF |
| $GLCM_{Var}$ [$\star$] | Variance of the GLCM | MOD/MYD02 | **NN**/RF |
| $GLCM_{Hom}$ [$\star$] | Homogeneity of the GLCM | MOD/MYD02 | NN/RF |
| | | | |
| $IST$ | Ice-Surface Temperature | MOD/MYD29 | TIT |
| | | | |
| $\sigma^0$ | Calibrated Backscatter | S1-A/B | R |
| | | | |
| $T2m$ | 2 m Temperature | ERA-Interim | TIT |
| $T_d2m$ | 2 m Dew-Point Temperature | ERA-Interim | TIT |
| $mslp$ | Mean Sea-Level Pressure | ERA-Interim | TIT |
| $u10m$ | 10 m u Wind Component | ERA-Interim | TIT |
| $v10m$ | 10 m v Wind Component | ERA-Interim | TIT |

AE=Autoencoder; NN=Neural Network; RF=Random Forest; R=Reference; TIT=Thin-Ice-Thickness Calculation

$\star$=Calculated/Derived for MODIS channels: 20, 22, 23, 24, 25, 29, 31, 32, 33, and 35;

$\bullet$=Calculated as difference (later minus earlier acquired swath) between temporal neighbors;

$\diamond$=Normalized through swath-wide mean and standard deviation: $BT_{norm} = (BT - \overline{BT}) \times \sigma_{BT}^{-1}$

approach. For visual reference, the S1-A/B data is also resampled to an equi-rectangular grid with the same extent but a spatial resolution of 25 m. Through the decreasing distance between meridians towards the pole, the per-pixel spatial area also decreases. This results from the constant latitudinal distance between grid points in this type of projection. Ice-shelf areas are

excluded from our analysis based on Rtopo-2 data (Schaffer et al., 2016).



### 2.2.1 MOD/MYD02 L1b calibrated radiances

Our goal for the later discrimination algorithm was for it to solely rely on MODIS-channel data, without the need for any auxiliary data. Brightness temperatures (BT) were calculated from calibrated radiances comprising MODIS channels 20, 22, 23, 24, 25, 29, 31, 32, 33, and 35 following Toller et al. (2009). This channel subset allows to distinguish between sea-ice,

open-water/thin-ice, and cloud pixels through a high inter-channel variability.

Furthermore, we computed additional parameters from these BT comprising ice-surface temperature (IST; following (Riggs and Hall, 2015) as well as image-texture parameters using GLCM (Tab. 1).

We generally limited our swaths to sensor incidence angles $\leq 60\,^\circ$ to minimize spatial distortion towards the swath edges and a coverage $>90\,\%$ of our reference area.

### 2.2.2 MOD/MYD29 sea-ice product

For a later comparison based on cloud coverage, polynya area, and sea-ice production rates, we extract and use the IST from the reference NSIDC sea-ice product, which offers an overall accuracy of 1–3 K under ideal (i.e., clear-sky) conditions (Hall et al., 2004; Riggs and Hall, 2015).

Both IST (MOD/MYD02 and MOD/MYD29) are derived based on a constant emissivity for snow/ice (Hall et al., 2015), but

with the MODIS cloud mask already applied to the MOD/MYD29 product.

### 2.2.3 S1-A/B L1 calibrated backscatter

In order to reliably identify polynyas independent of cloud-cover or other atmospheric disturbances, we selected a total of eight S1-A/B swaths featuring an active polynya in front of the BIS.

These S1-A/B swaths together with co-located and at least partially cloud-free MOD/MYD02 data are used for training and

validation of the algorithm. S1-A/B swath acquisition times are temporarily distributed over the 2017 Antarctic winter, with all additional information summarized in Table 2.

### 2.2.4 ERA-Interim data and thin-ice retrieval

For a quantitative comparison between resulting polynya area (i.e., the total area of pixels covered with a maximum ice thickness of 0.2 m), we calculate the thin-ice thickness (TIT) from MODIS IST for MOD/MYD02 und MOD/MYD29 data using a

surface-energy-balance model together with the ERA-Interim 2 m air temperature, the 10 m wind-speed components, the mean sea-level pressure, and the 2 m dew-point temperature (Dee et al., 2011).

The surface-energy-balance model utilizes the relation between IST and the thickness of thin sea ice (Yu and Rothrock, 1996; Drucker et al., 2003).Generally, thinner ice features a higher IST than thicker ice as the influence from the warm ocean is diminished. The net positive flux towards the atmosphere between the warm ocean and the cold atmosphere is equalized from

the conductive heat flux through the ice. From the conductive heat flux the TIT is derived. A detailed description of the retrieval





**Table 2.** List of used S1-A/B swaths for training and validation/analysis (Fig. 6).

| Satellite | Product | Acquisition in UTC |
|---|---|---|
| | Training | |
| S1-A | EW_GRDM_1SSH | 2017-06-21 23:23:14 |
| S1-A | EW_GRDM_1SSH | 2017-07-08 23:31:21 |
| S1-A | IW_GRDH_1SSH | 2017-07-13 03:50:25 |
| S1-A | IW_GRDH_1SSH | 2017-08-06 03:50:27 |
| S1-B | IW_GRDH_1SSH | 2017-08-19 23:30:41 |
| S1-A | EW_GRDM_1SSH | 2017-09-20 00:11:57 |
| | Validation | |
| S1-B | IW_GRDH_1SSH | 2017-04-02 03:49:42 |
| S1-A | EW_GRDM_1SSH | 2017-04-07 22:58:51 |
| S1-B | EW_GRDM_1SSH | 2017-04-09 00:27:24 |
| S1-A | EW_GRDM_1SSH | 2017-05-11 00:11:50 |
| S1-A | EW_GRDM_1SSH | 2017-05-16 23:23:12 |
| S1-A | EW_GRDM_1SSH | 2017-05-18 23:06:59 |
| S1-A | EW_GRDM_1SSH | 2017-07-20 00:28:11 |
| S1-A | IW_GRDH_1SSH | 2017-09-11 03:50:28 |

procedure as well as all equations and necessary assumptions are thoroughly described in Paul et al. (2015) as well as Adams et al. (2013). For ice thicknesses between $0.0\,\text{m}$ and $0.2\,\text{m}$, Adams et al. (2013) state an average uncertainty of $\pm 4.7\,\text{cm}$.

## 2.3 Initial training data generation

The availability and quality of labeled training data is of utmost importance for the training of any supervised machine-learning algorithm. However, available spatio-temporal high-resolution cloud information over nighttime sea ice is practically non-existent. Therefore, we had to derive our own labeled training data using co-located MODIS and S1-A/B data to manually identify cloud, sea-ice as well as open-water/thin-ice pixels, respectively.

To reduce manual effort and uncertainty to a minimum, we employ a mix of dimension reduction and unsupervised clustering before the final manual classification.

First, we selected up to 13 MODIS swaths in close temporal proximity for each of the six S1-A/B reference swaths (Top six in Tab. 2), i.e. in a temporal range of $\pm 2\,\text{d}$ around the S1-A/B swath. Secondly, in addition to the textural parameters from the GLCM (Tab. 1), we wanted to add a temporal component to the parameter mix. We, therefore, added the IST of two swaths acquired before and after the current swath, respectively.

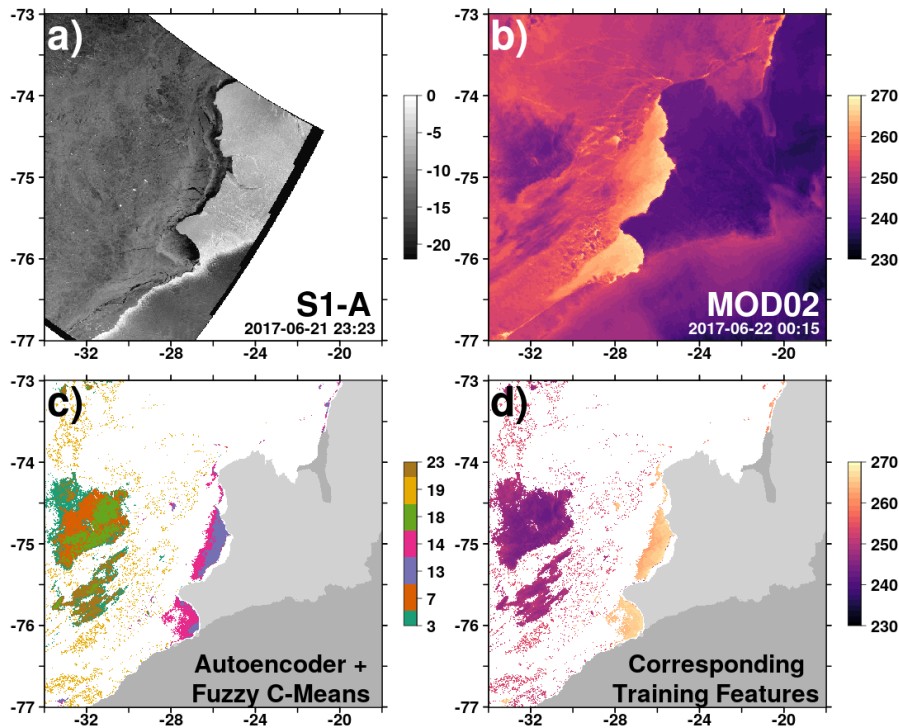

**Figure 3.** Examplary generation of labeled training data with reference Sentinel-1 A/B calibrated backscatter image (in dB; a), MOD02 derived ice-surface temperature (in K; b), subset of exemplary results from the used Autoencoder and Fuzzy C-Means Clustering (c), and the corresponding ice-surface temperature of the selected example cluster subsets (in K; d). Here, clusters 3, 7, 18, and 23 were categorized as 'cloud'; clusters 13 and 14 as 'open-water/thin-ice'; and cluster 19 as 'sea ice'. Land-ice (dark gray) and ice-shelf (light gray) overlays originate from Rtopo-2 (Schaffer et al., 2016).

From here, we take advantage of the AE dimension reduction capabilities (Fig. 2). Hence, instead of using the total number of
39 input parameters for the FCM with probably only mediocre results, we cluster the encoded information from the three bottle-neck layer neurons for all pre-selected and pre-processed MODIS swaths. Subsequently, the FCM soft-clusters similar pixels into 35 clusters before we manually categorize these clusters into one of the three classes 'cloud', 'sea ice', or 'open water/thin ice'. An exemplary sequence of this procedure is shown in Figure 3. In order to reduce uncertainty in the training data, we constrained the manual classification to 'obvious' cases (e.g. 'cold' continuous patches over otherwise 'warm' polynyas and
adjacent sea-ice categorized as 'clouds'), which results in not every MOD/MYD02 swath being fully classified at this stage.

We used a subset ($n$=11) of all available co-located MODIS swaths ($n$=52) in order to i) keep the manual categorization computationally and humanly manageable and ii) represent easily distinguishable configurations of cloud, sea-ice and open-water/thin-ice pixels with favorable geometries (e.g., low sensor zenith angles). To increase the variability of the feature space in our labeled training data set, we selected the neighboring four swaths randomly from a pool of MOD/MYD02 swaths with
a maximum time difference of $\pm 24$ h to the current MODIS swath. However, we manually classified two combinations of the




randomly selected temporal neighbors and applied these results to up to 25 random combinations for each of the 11 swaths. This increased the overall temporal variability of our training data set substantially.

Finally, from our manual categorization, we only use pixels with a FCM probability (i.e., the membership score) above 0.6 for 'open-water/thin-ice' pixels, 0.65 for 'sea-ice' pixels as well as 0.65 for 'cloud' pixels. As 'sea-ice'/'cloud' pixels are

harder to identify, we chose a stricter probability threshold for those two classes. Due to the large temperature range present in Antarctic clouds, we separated our 'cloud' class internally into 'cold' ($<235\,\mathrm{K}$), 'intermediate', and 'warm' ($>250\,\mathrm{K}$) clouds. This separation leads to an improved general classification result for the supervised learning algorithms. All ice-shelf areas are excluded from our analysis to avoid any additional misclassifications due to the substantially different temperature regime.

Through this procedure, we created an initial labeled training data set consisting of about $8.2\times10^6$ data points.

## 215  2.4  Training of classifier candidates

While some general rules of thumb exist for the setup of NN and RF, the process of finding the best setup is iteratively based on training with varying setups and subsequent assessing of the resulting accuracy based on validation data. Therefore, before training any classifier, we separated our $8.2\times10^6$ labeled training data points into a calibration and a validation part using randomly selected 75 %/25 % of the total data points, respectively (Fig. 2).

For classifier-candidate training, we use the following variations for the setup of our NN:

– the activation functions (ReLU/TanH; either constant for all hidden layers in a NN or combinations of them),

– the number of hidden layers (2-4) as well as the number of neurons (10, 20 or 25; either constant for all layers or decreasing per layer) in each hidden layer, and

– the number of training epochs (10-50).

For the setup of the RF classifiers we change:

– the number of trees in each RF (25-40); while at the same time

– using a constant $m$ of 3 for selection of split parameters during decision-tree growth.

For both classifiers, we varied and limited the input parameters for both classifier types (Tab. 1).

## 2.5  Validation and final training data generation

From assessing the classification accuracy on the validation data set as well as visual screening it became apparent that all of the classifier candidates showed biases towards either one or several of 'open-water/thin-ice', 'sea-ice' or 'cloud' classes. This likely results from either a higher degree of complexity in the data that cannot be represented in the current classifier setups or a lack of representation of the overall variability in the data.

In order to increase our training data sample size to fully represent more different configurations of 'cloud', 'sea-ice', and

'open-water/thin-ice' pixels, we selected eight classifier candidates with the highest accuracy (5×NN, 3×RF) and used them



**Table 3.** Normalized confusion matrix for the final deep NN classifier between reference (R; columns) and prediction (P; rows) of the NN for the five classes of open-water/thin-ice pixels (OWT), sea-ice pixels (ICE) as well as the three temperature-dependent cloud sub-classes (CLD); (w)arm, (i)ntermediate, and (c)old.

| ↓ P \R → | OWT | $CLD_c$ | $CLD_i$ | $CLD_w$ | ICE |
|---|---|---|---|---|---|
| OWT | **0.95** | 0.00 | 0.00 | 0.01 | 0.01 |
| $CLD_c$ | 0.00 | **0.93** | 0.00 | 0.00 | 0.00 |
| $CLD_i$ | 0.00 | 0.05 | **0.95** | 0.02 | 0.04 |
| $CLD_w$ | 0.02 | 0.00 | 0.01 | **0.91** | 0.03 |
| ICE | 0.03 | 0.02 | 0.04 | 0.06 | **0.92** |

to classify all of our co-located MODIS swaths ($n$= 52). Here, we randomly selected three temporal neighbor combinations for each of the 52 swaths.

In this way, we increased our total variability in the training data (while keeping it computationally manageable), and got a statistically substantiated classification of the complete swaths - in contrast to the partially categorized swaths through manual classification used before. Furthermore, we reduced the number of input predictors to 45 by removing consistently less important predictors through using the RF importance metric (Tab. 1).

An ensemble combination of the best classifier results was used for the final training data, based on the probability estimates provided by the RF/NN classifiers as well as again manual visual screening.

All RF classifiers featured frequent rather close decisions in their resulting classifications, i.e. mid range probabilities. This likely originated from comparably few trees in the RF classifiers. However, larger numbers of tree could not be realized with the given computing power.

Through this procedure, we created our final labeled training data set of about $17.0 \times 10^6$ data points comprising 45 different predictors/parameters (Tab. 1).

## 2.6 Training of the final classifier

This final training data set we again randomly split into calibration/validation data using 75 %/25 % of the total data points, respectively (Fig. 2).

As we could not increase the number of trees in the RF classifiers due to computing limitations, we settled for a NN as our final classifier type. All final classifier candidates are using ReLU activation functions only, logarithmic loss function, a batch size of 100, and a total of 15 training epochs.

The only parameter we change for assessing the accuracy between different classifier candidates is the number of fully-coupled hidden layers (4-8), with each consisting of 20 neurons per hidden layer.





**Table 4.** Summary of polynya area (PA; in $km^2$) estimates between S1-A/B ($PA_{S1}$), OSCD ($PA_{OSC}$), and MOD/MYD29 ($PA_{M29}$) data. PA estimates in parenthesis correspond to the PA retrieved from MODIS for the S1-A/B polygon.

| Example | $PA_{S1}$ | $PA_{OSC}$ | $PA_{M29}$ |
|---|---|---|---|
| Fig. 4a-d | 4235 | 6472 (4074) | 5134 (1823) |
| Fig. 4e-h | 2224 | 2586 (2129) | 3146 (2131) |
| Fig. 4i-l | 380 | 484 (366) | 47 (18) |
| Fig. 5a-d | 1093 | 1037 (880) | 962 (920) |
| Fig. 5e-h | 1448 | 1314 (1220) | 730 (722) |
| Fig. 5i-l | 1425 | 2172 (1414) | 1019 (257) |

The best result was achieved with a setup of six hidden layers (Table 3), featuring an overall accuracy of 93.2 %, i.e. the ratio of correctly classified pixels (3,961,579) to the total number of samples (4,252,193) in the validation data set. For our comparisons and the results, we always merged all cloud sub-classes to a single cloud class.

## 3 Results and Discussion

In the following, we describe and discuss the results using the final NN classifier for our open-water/thin-ice, sea-ice, cloud discrimination (OSCD) product in comparison to the reference MOD/MYD29 sea-ice product.

Representative comparisons between resulting thin-ice thickness (TIT) from OSCD and MOD/MYD29 swaths reveal substantial differences, especially in the high-temperature polynya and thin-ice areas (PA; Figs. 4 & 5).

The S1-A/B reference data always feature a polynya signal in all our examples (Figs. 4a,e,i & 5a,e,i) and these are at least partially represented by a warm IST anomaly in the MODIS data (Figs. 4b,f,j & 5b,f,j). While for some examples the difference in resulting TIT between OSCD and MOD/MYD29 is comparably small or negliable (Figs. 4g/h & 5c/d), substantial differences appear for other examples (Figs. 4c,k/4d,l & 5g,k/5h,l).

For a better comparison, the polynyas were hand-picked for S1-A/B and MODIS data in Figures 4 & 5. The corresponding absolute polynya areas are summarized in Table 4. In addition to the corresponding numbers for each polynya, the corresponding area covered in the S1-A/B extent is given in parenthesis. While there is a bit of uncertainty due to the different grid resolution (25 m vs. 1 km), this allows for a good quantification of the impact of misclassified cloud cover.

While there are correct and also corresponding cloud classifications in both MODIS products, the applied MODIS cloud mask in the MOD/MYD29 product tends towards additionally masking out strong positive temperature anomalies (Figs. 4d & 5h,l). This happens frequently in the center of the primary polynya around 27.4 °W and 76 °S and leads to substantial differences in PA estimates (Tab. 4).

Due to the strong temperature gradient between the warm ocean and the cold atmosphere, turbulent exchange of sensible and latent heat is large and can potentially lead to the formation of sea fog and thin, low cloud cover (Gultepe et al., 2003).



**Figure 4.** Compilation of exemplary co-located S1-A/B calibrated backscatter (in dB) and MODIS swaths of ice-surface temperature (IST; in K) and derived thin-ice thickness (TIT; in m) data (Tab. 2). Gray/green overlays highlight the ice-shelf extent. Manually picked S1-A/B reference polynya extent is outlined by a dashed red line.

However, the temperature texture in the open-water/thin-ice areas appear to be homogeneous, and is likely not to be affected
by either sea fog or clouds to the extent suggested by the MOD/MYD29 product through the MODIS cloud mask.





**Figure 5.** Additional compilation of exemplary co-located S1-A/B and MODIS swaths in the same setup as Figure 4. Gray/green overlays highlight the ice-shelf extent. Manually picked S1-A/B reference polynya extent is outlined by a dashed red line in all panels.

Based on median TIT of all available MODIS swaths per day, daily polynya area (PA) was calculated, and the difference between OSCD and MOD/MYD29 estimated (i.e. OSCD minus MOD/MYD29; Fig. 6).





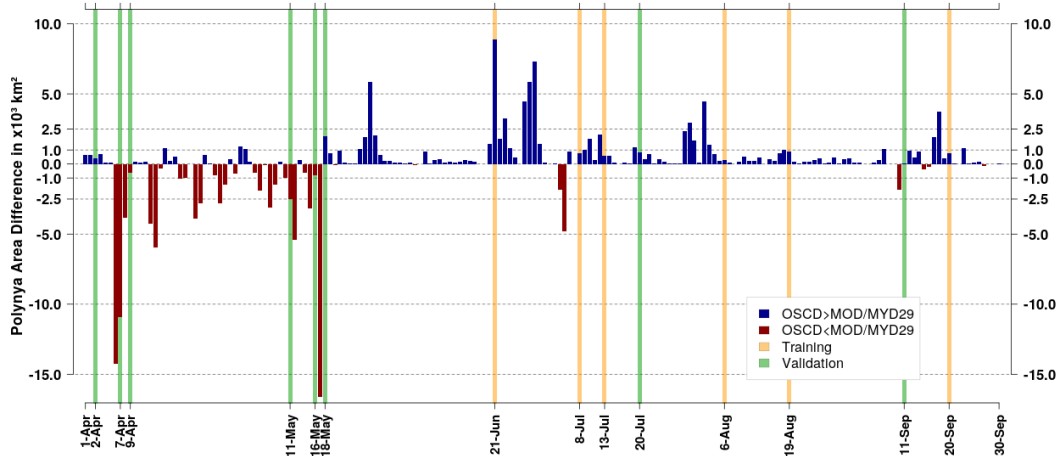

**Figure 6.** Daily polynya area difference in $\times 10^3\,\mathrm{km}^2$ using swath-wise pixel averages featuring a thin-ice thickness (TIT) $\leq 0.2\,\mathrm{m}$ between OSCD and MOD/MYD29. Difference is calculated by subtracting MOD/MYD29 from OSCD; results with OSCD$\geq$MOD/MYD29 are shown in blue; results with OSCD$<$MOD/MYD29 in red. Orange vertical bars highlight days with S1-A/B swath coverage used for training of the OSCD algorithm. Green vertical bars show additional S1-A/B swaths used for validation between products. Additional information about the S1-A/B swaths is provided in Table 2.

Larger estimates in MOD/MYD29 appear to dominate from 1 April 2017 to mid May 2017, while OSCD estimates are in general larger or equal to MOD/MYD29 between mid May and 30 September 2017. For the year 2017, about 72.3 %/56.0 %/26.6 %

of the absolute daily median PA differences are below $1000\,\mathrm{km}^2$/$500\,\mathrm{km}^2$/$100\,\mathrm{km}^2$, respectively.

On average, OSCD estimates the daily polynya area (PA) between 1 April and 30 September 2017 to be $2.79 \times 10^3\,\mathrm{km}^2$ in contrast to $2.75 \times 10^3\,\mathrm{km}^2$ using MOD/MYD29 data. This corresponds to an average of 2 % smaller daily mean PA for MOD/MYD29. From our comparisons based on swath data and the sometimes substantial differences we have found (Figs. 4 & 5), this appears to be a very small difference. An explanation could be that the composites counterbalance the individual

shortcomings in the swath data from misclassification of thin ice as cloud cover in the MOD/MYD29 product.

However, especially during freeze-up (i.e. between 1 April 2017 and mid May 2017), the differences are oftentimes very large ($16.6 \times 10^3\,\mathrm{km}^2$ on 17 May 2017) and towards MOD/MYD29. To analyze this, we want to conduct a more detailed analysis of OSCD and MOD/MYD29 daily median TIT (Figs. 7 & 8).

Unfortunately, no S1-A/B swath was acquired over the BIS area for 17 May 2017. However, S1-A/B swaths were acquired

the day before and after (Tab. 2 and Fig. 6).

From the S1-A/B data (Fig. 7a/b), the existence of open water and/or thin ice very close to the ice-shelf edge around $27.4\,^\circ\mathrm{W}$ and $76\,^\circ\mathrm{S}$ for 18 May 2017 is evident.

The lack of any distinguishable features in the MODIS daily median IST composite (Fig. 7c) as well as the general texture of rather smooth temperature gradients are both signs for a persistently present cloud cover during 17 May 2017.



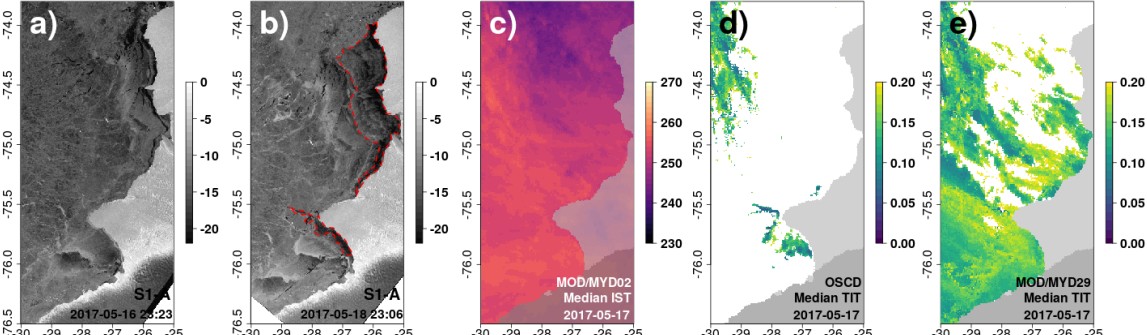

**Figure 7.** Compilation of S1-A swaths acquired at 16/18 May 2017 (a/b; calibrated backscatter in dB), the daily median ice-surface temperature (IST; in K) composite of 17 May 2017 from all available MODIS swaths (c), as well as the resulting daily median thin-ice thickness (TIT; in m) composites for the OSCD (d) and MOD/MYD29 (e) products for 17 May 2017, respectively. Red dashed line outlines the polynya on 18 May 2017 in S1-A.

However, the relatively high temperatures of these potential clouds lead to an erroneous calculation of TIT and subsequent daily median TIT composite with an erroneously much larger polynya area (PA) for MOD/MYD29 compared to OSCD (Fig. 7d/e). Nonetheless, also OSCD features TIT estimates from cloud artifacts in the NW around 29.5 °W and 74-74.5 °S.

  The individual swaths used for the computation of the composites underline the absence of any pronounced positive temperature anomalies corresponding to open-water/thin-ice features (Fig. 8a-h).

The inability of the MODIS cloud mask to reliably identify these cloud patterns results in the computation of TIT in large patches West of BIS (Fig. 8q-x). Conversely, these false computations are not present in the OSCD data (Fig. 8i-p). However, while a small area West of the tip of BIS around 28 °W and 75.5 °S corresponds well to the polynya signal in the S1-A data (Fig. 7b), the majority of TIT estimates appear to be cloud artifacts (Fig. 8p).

  Two reasons can, therefore, explain the good overall agreement in daily median PA values:

1. Erroneous TIT estimates due to cloud-cover artifacts in the MOD/MYD29 data increase the overall estimated PA; and

  2. Over the course of one day, the frequent MODIS coverage as well as the rapid growth rate of very thin ice potentially leads to rather sufficient coverage in the MOD/MYD29 product.

This combined effect leads to spatially misplaced TIT estimates, likely not resolving the correct shape and (sub-)daily thickness distribution of the open-water/thin-ice areas.

While the annual average daily PA appears to be consistent between OSCD and MOD/MYD29 data, per-swath coverage of thin ice is much higher in the OSCD data compared to the MOD/MYD29 data (Fig 9).

  Especially in the very active part of the polynya with frequent thin-ice occurrences around 27 °W and 76 °S, OSCD produces a much higher detection-rate of thin-ice pixels over all MODIS swaths (Fig 9b).

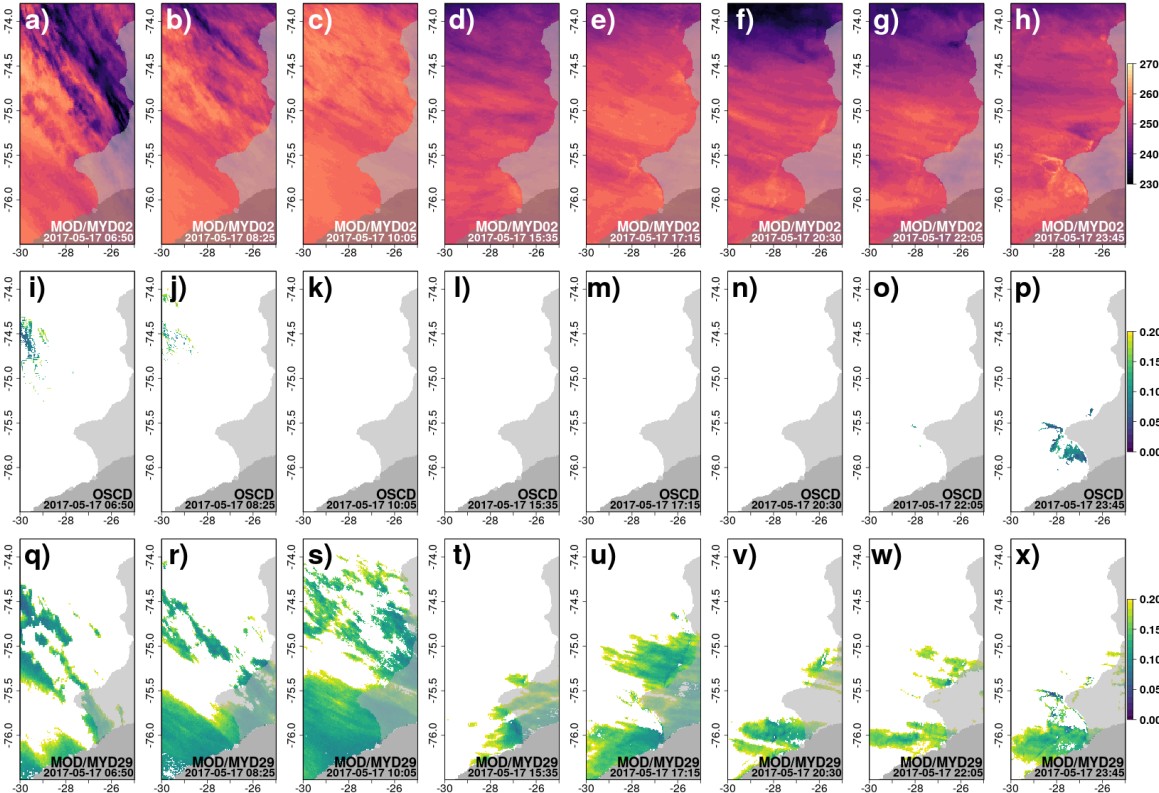

**Figure 8.** Compilation of MODIS swaths used for the computation of the data shown in Figure 7: swath-based ice-surface temperature (IST in K; a-h), resulting swath-wise thin-ice thickness (TIT in m) using OSCD (i-p) and MOD/MYD29 (q-x) data, respectively. Due to space limitations, we excludes a total of four MODIS swaths in this Figure that were used in the daily computations.

For the core polynya region (Green dashed outline; Fig. 9), use of OSCD results in a substantial 31 % increase in coverage
between 1 April 2017 and 30 September 2017. The improved coverage likely leads to a higher quality daily composite, as the impact from outliers is reduced. This will likely play an important role in the estimation of sea-ice production.

Despite great care during the manual categorization, uncertainty remains due to the lack of measured ground-truth data for the training-data generation.

However, the underlying statistical basis from the unsupervised FCM clustering in combination with a second stage of fully
classifying all co-located MODIS swaths using RF/NN appears to provide a realistic representation of the present sea-ice conditions in the BIS area.




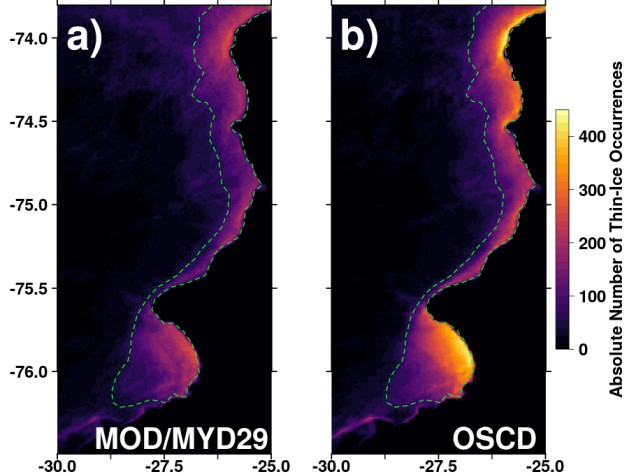

**Figure 9.** Comparison of per-pixel thin-ice occurrence based on all available swaths from 1 April to 30 September 2017 between the use of MOD/MYD29 (a) and OSCD data (b), respectively. Green dashed line marks the core polynya region used for analysis in the text.

## 4 Summary and Outlook

In this study, we present a novel approach to improve the detection of wintertime cloud-cover over Antarctic sea ice and their discrimination from sea-ice cover and open-water/thin-ice areas in MODIS thermal-infrared data using a deep neural network.

We established a labeled training data set using dimension reduction, unsupervised clustering, and supervised learning techniques in combination with manual visual screening and categorization. Through this effort, we generated a total of $17.0 \times 10^6$ data points for 45 different predictors.

With this data set, we trained a deep neural network and used it to discriminate between open-water/thin-ice, sea-ice and cloud-covered areas in the Brunt Ice Shelf region for the freezing period of 2017 (1 April to 30 September). Here, we computed

the thin-ice thickness up to $0.2\,\mathrm{m}$ of open-water/thin-ice areas and evaluated the difference in daily polynya area and daily swath coverage to results using the standard NSIDC MOD/MYD29 sea-ice product.

Based on our approach, we obtain a 2 % higher average polynya area but 31 % higher swath coverage rate. However, the polynya area in MOD/MYD29 is likely dominated through frequent misclassifications of warm clouds as thin ice, that lead to unrealistically large open-water/thin-ice areas, especially during freeze-up. The much higher coverage rate likely increases

the quality and accuracy of TIT estimates in the daily median TIT composites when using OSCD data. This also reduces the impact of single outliers on the daily median TIT composites and, therefore, also increases the quality of derived information such as sea-ice production.

In the future, we plan to create an open-access comprehensive OSCD-based IST/TIT product covering all major Antarctic coastal polynyas, as well as providing higher-level parameters such as polynya area, sea-ice production, and associated ocean





salt flux. We expect this data set to be of great use to ocean/sea-ice/ice-shelf model community as well as for potential biological applications.

*Data availability.* The generated training data set will be made available through PANGAEA. Sources of all used data sets are referenced in the text.

*Author contributions.* SP designed the study/methodology, conducted the analysis, and drafted the original manuscript. MH assisted in the
study design and the adaptation of the machine learning algorithms as well as with writing the manuscript.

*Competing interests.* The authors declare no conflict of interests.

*Acknowledgements.* The authors want to thank the LAADS DAAC and the ASF DAAC for the provision of the here used MOD/MYD02 and S1-A/B data. The corresponding author appreciates the help of his family in enabling him the time to finally write this manuscript during SARS-CoV-2 induced home office.



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
