# Peer review of "Improved machine-learning based open-water/sea-ice/cloud discrimination over wintertime Antarctic sea ice using MODIS thermal-infrared imagery"

_The Cryosphere, 2020_

## Referee Comment (RC1) · Anonymous Referee #1 · 3 Aug 2020

A review of tc-2020-159, by Stephen Paul and Marcus Huntemann

I have completed a review of this manuscript and recommend publication after minor revisions, most of which are just for clarification.

In this paper, the authors present a significant advance in the accurate detection of cloud in polar regions using advanced machine-learning techniques, and go on to illustrate the utility of this algorithm by retrieving much better estimates of thin ice thickness in the vicinity of the Brunt Ice Shelf, West Antarctica. The novel approach and subse-

quent advances in cloud detection are sorely needed for polar MODIS users, and as such I believe this paper is highly suitable for publication in TC, after minor modifications outlined below.

L22-29: It would be nice to point to other polar MODIS applications which would benefit from such a better cloud mask. Examples include composite image generation, landfast ice mapping, possibly sea ice motion retrieval using image cross-correlation.

L29: This problem has also been reported in coastal leads, e.g. Fig. 6 from Fraser et al., 2009. DOI: 10.1109/TGRS.2009.2019726

L40: Why was this study region chosen? Is this applicable for both flaw leads and nonlinear coastal polynyas (e.g., Terranova Bay)?

L54: At this stage, It strikes me that it might be better to describe input data before discussing the methods (i.e., move section 2.2 to here). This may have just been my personal preference though.

L112: This parapgraph needs more explanation. E.g., the 29-23... metric. Also, why are some numbers in bold type? What are the 35 epochs? 100 whats in a batch? Huber needs a capital H too.

Section 2.2: You need to describe the version of the MODIS products – particularly the MOD29 product. This determines which MOD35 version went into the product. There have been some significant improvements to MOD35 over the years, so it's important to document which version.

There is no description of the gridding or MODIS destriping/de-bowtie here. These must have huge influence on the performance of the algorithm, so a description of these processes is needed, in my opinion. Many of the channels used suffer strongly from detector striping in particular.

L140: What is the resolution of the IST product?

Section 2.2.1: The destriping description may fit better here.

L158: What about the increased atmospheric path encountered for high incidence angles – is that more important than the geometry distortion?

L164: Does MOD/MYD29 also apply atmospheric correction to more accurately determine IST?

L167: I guess you're developing this algorithm for coastal, latent heat polynyas. It might be good to make this clear here. I doubt it would work for offshore/sensible heat ones (which is fine)!

L178: This sentence "Generally, " is somewhat ambiguous.

Fig 3: One sub-figure would benefit from including a distance scale.

Fig 3: "Examplary" typo.

L219: The cal/val split was done on a point-wise basis? This seems a bit strange. Isn't the point of the cal/val split to ensure independence between the calibration and validation datasets by withholding at a more basic level, e.g., scene level? What I'm trying to say is, two neighbouring pixels are unlikely to be completely independent. So if there's a 75% chance of each pixel getting into the training dataset, then it's pretty much guaranteed that the validation dataset won't be particularly independent of the training dataset. Could you comment on this?

L278: Again, the Fraser et al., 2009 reference which shows this in a flaw lead would be good to reference here.

Fig 4, 5: There is a mismatch between the actual extent of the Brunt Ice Shelf and the masked version, based on the Rtopo product. This is due to ice shelf advection in the time between the creation of both products. In this case, there are both areas of ice shelf outside the mask, and areas of water/sea ice within the mask. Other high-resolution coastal datasets have mitigated this by including a manually-updated ice

shelf extent product on a regular basis (e.g., Fraser et al., 2020, ESSD Discussions, https://doi.org/10.5194/essd-2020-99), but this level of mitigation is probably unwarranted here. However, could you comment on the effect this might have on the training algorithm?

Fig 6: This is a great way of showing the seasonality in bias. However I'm still hanging out for a good old-fashioned scatterplot comparing these two datasets. This would show highly complementary information to your time series.

L286: I think the "average" metric you use here may not be the best way to highlight how much better your algorithm performs! Have you considered also using RMS difference?

L303: Unclear which product that this statement corresponds to.

L309: "Good agreement" between what and what?

L337: Again, the suggested RMS statistic would better highlight your improvement.

---

## Referee Comment (RC2) · Anonymous Referee #2 · 6 Nov 2020

This paper tried to detect a cloud-covered area from a satellite thermal infrared image based on the mechanical-learning. The analysis area is a coastal polynya area close to the Brunt Ice Shelf, Antarctica. A coastal polynya is a low ice concentration or thin ice-covered area formed by divergent ice motion due to wind or ocean current. Heat insulation effect by sea ice is reduced significantly in the case of thin ice or open water fraction. Therefore, large heat loss from the ocean to the atmosphere occurs in a winter coastal polynya, leading to active freezing. The resultant large amount of cold and saline water (brine) rejection leads to the formation of dense water, which is a major

source of the bottom water. The sinking of the dense water plays a significant role in the global climate system by driving thermohaline circulation and exchanging $CO_2$ between the atmosphere and the deep ocean. Thus, the Antarctic coastal polynyas are vital areas for the global climate system despite their relatively small areal extent (100 km at most).

Satellite data observed by passive microwave (PM) sensor (e.g., SSM/I, SSMIS, AMSR-E/2) are often used for sea-ice monitoring because the sensor can observe the earth surface regardless of darkness and cloud cover. The PM data are also used for coastal polynya studies, and a map of coastal polynyas of the hemispheric scale is revealed (e.g., Tamura et al., 2008; Nihashi and Ohshima, 2015; Nihashi et al., 2018). The disadvantage of PM data is the coarse spatial resolution of 6-12 km. On the other hand, the spatial resolution of a thermal infrared image observed by MODIS used in this manuscript is about 1 km. This higher spatial resolution allows for more detailed monitoring of a coastal polynya. A weak point of the thermal infrared image is that the sensor cannot observe the earth's surface due to darkness and cloud cover, different from the PM sensor. As described in the manuscript, the MODIS data includes cloud information (mask). This cloud mask works well at the normal sea-ice covered area where the ice thickness is not thin. Air temperature over a coastal polynya area is relatively warm due to heat flux from the ocean. This leads to errors in the MODIS cloud mask, and unfortunately, the mask virtually cannot be used over coastal polynya areas. The thin ice thickness algorithm development using PM data is based on comparisons with thermal ice thickness derived from MODIS infrared images (Nihashi and Ohshima, 2015). They found out cloud-free infrared images by manual inspection. I presume that the manuscript's findings can help this algorithm development, and this also would be a useful application. Referring to such a study might raise the value of this manuscript.

In the field of remote sensing, research applying machine learning has been increasing rapidly in recent years. Nevertheless, it is not easy to physically interpret machine learning results in geophysical ones, and many of them are not successful. On the

other hand, this manuscript's results seem to be an excellent example of applying machine learning to the field of sea-ice remote sensing. The manuscript is well prepared. Perhaps readers interested in this content can predict that they are not very knowledgeable about machine learning, including me, although machine learning experts may not be very interested in this paper. Even for such a beginner, this manuscript explained machine learning clearly and concisely, and I felt it was terrific. Based on my review, I recommend this manuscript for publication in the Cryosphere after minor revision.

Major comments:

There are many coastal polynyas around Antarctica. A map of sea-ice production in coastal polynyas derived from the PM data (Nihahshi and Ohshima, 2015; Nihashi et al., 2018) shows that the ice production in a coastal polynya near the Brunt Ice Shelf region is small. This indicates that this small polynya's impact on climate change is not so significant as the other larger polynyas. Why did the authors focus on this minor polynya as a study area? In the title and throughout the manuscript, as stated in "Antarctic sea ice", it gives the impression that this manuscript's results have been achieved as if they represent the entire Antarctic Ocean. I wonder that a result of the cloud mask from a small polynya study can represent the entire Antarctic coastal polynyas or that the result of this manuscript can be applied to other large major coastal polynyas, such as the Ross Ice Shelf Polynya.

Minor comments:

P. 2, L. 22: Please correct "polyanya" to "polynya".

P. 2, L. 23: "both in, the Arctic ...". It would be "both in the Arctic ...".

P. 3, Fig. 2: There are no linkages between ted characters of A-H and the manuscript. I felt that elaborating by following these in the manuscript would be helpful for readers.

P. 4, L. 67: "ii) large number". It would be "ii) a large number".

P. 5, L. 103: "hyberbolic tangent". It should be "hyperbolic tangent".

P. 5, L. 112: Please explain the number: "39-23-10-3-10-23-39" and the meaning of the number shown by the bold character of "**39-23-10-3**".

P. 8, L. 178: Please correct from ", 2003).Generally" to ", 2003). Generally". Insert a space.

P.9, L. 184: "... data is of ...". It would be "...data are of ...".

P. 10, Fig. 3c: Correspondence between cloud, open water/thin ice, and sea ice and color is hard to identify. For example, my suggestion is that clusters 3, 7, 18, and 23 that correspond to the cloud area are shown by similar colors that can clearly distinguish from open water/thin ice and sea ice areas. Also, do each of the 3, 7, 18, and 23 clusters that correspond to clouds reflect the type of cloud? Further, what is the white area that does not belong to any cluster in this figure?

P.11, L. 208: "... a FCM probability ...". It would be "... an FCM probability ...".

P. 11, L. 212: The authors defined threshold values of temperature. How did you define these values? Is there any physical background?

P. 13, L. 267: "negliable". It should be "negligible".

P. 14, Fig. 4a, e, and i: A polynya area surrounded by red line: the authors described that the area was "manually picked". How did you define the polynya area?

P. 15, Fig. 5a, e, and i: Same as the above.

P. 15, bottom: "and MOD/MYD29 estimated". It would be "and MOD/MYD29 was estimated".

P. 17, L. 307: "... West of ...". It would be "... west of ...".

—

References:

Nihashi, S. and K. I. Ohshima (2015), Circumpolar mapping of Antarctic coastal polynyas and landfast sea ice: relationship and variability, J. Clim., 28 3650–3670.

Nihashi, S., K. I. Ohshima, and T. Tamura (2017), Sea-Ice Production in Antarctic Coastal Polynyas Estimated from AMSR2 Data and Its Validation Using AMSR-E and SSM/I-SSMIS Data, IEEE JSTARS, 10(9), 3912–3922.

―――――――――――――――――――――

---

## Author Comment (AC1) · 2 Dec 2020

The very helpful comments and suggestions by Reviewer 1 are very much appreciated and we would like to thank him/her for the time and effort he/she put into this review of our manuscript. In the following, we would like to go through all comments/suggestions and reply to them or answer them point-by-point. Reviewer comments are put in **bold font**, our replies are colored and changes inserted to the manuscript are put in *italics*.

**In this paper, the authors present a significant advance in the accurate detection of cloud in polar regions using advanced machine-learning techniques, and go on to illustrate the utility of this algorithm by retrieving much better estimates of thin ice thickness in the vicinity of the Brunt Ice Shelf, West Antarctica. The novel approach and subsequent advances in cloud detection are sorely needed for polar MODIS users, and as such I believe this paper is highly suitable for publication in TC, after minor modifications outlined below.**

We appreciate the comment and are happy to reply to the made comments.

**L22-29: It would be nice to point to other polar MODIS applications which would benefit from such a better cloud mask. Examples include composite image generation, landfast ice mapping, possibly sea ice motion retrieval using image cross-correlation.**

We will follow this suggestion and add both previous studies that could have benefited as well as future potential applications. Examples include:

Ludwig, V., G. Spreen, C. Haas, L. Istomina, F. Kauker, & D. Murashkin (2019). The 2018 North Greenland polynya observed by a newly introduced merged optical and passive microwave sea-ice concentration dataset. The Cryosphere, 13(7), 2051–2073. doi:10.5194/tc-13-2051-2019

Reiser, F.; Willmes, S.; Heinemann, G. A New Algorithm for Daily Sea Ice Lead Identification in the Arctic and Antarctic Winter from Thermal-Infrared Satellite Imagery. Remote Sens, 2020, 12, 1957. https://doi.org/10.3390/rs12121957

Aulicino, G.; Sansiviero, M.; Paul, S.; Cesarano, C.; Fusco, G.; Wadhams, P.; Budillon, G. A New Approach for Monitoring the Terra Nova Bay Polynya through MODIS Ice Surface Temperature Imagery and Its Validation during 2010 and 2011 Winter Seasons. Remote Sens. 2018, 10, 366.

**L29: This problem has also been reported in coastal leads, e.g. Fig. 6 from Fraser et al., 2009. DOI: 10.1109/TGRS.2009.2019726**

We will add the suggested reference-

**L40: Why was this study region chosen? Is this applicable for both flaw leads and nonlinear coastal polynyas (e.g., Terranova Bay)?**

This point was brought up by both reviewers. The region was chosen because of earlier experience with the region through the corresponding author (see Paul et al. 2015a,b) as well as several benefits for the setup of the algorithm we would like to explain in the following. While this polynya is not a major player in, e.g., deep-water formation (as pointed out by Reviewer 2) it is one of the most active regions in the Weddell Sea and similar in ice production to the much larger Ronne Ice Shelf polynya (see Paul et al. 2015b). The high activity as well as the good spatio-temporal coverage through Sentinel-1 led to

the decision to start-off with this region. However, the approach is assumed independent of the selected region as it mainly depends on the received satellite TIR signals, i.e., the temperature differences between surface types and clouds. Due to the investigation of a complete freezing period, this is assumed to be comparable to other polynya regions and should also be independent of polynya shape/size. A study applying the proposed procedure to all Antarctic polynyas is currently under preparation.

We will a clarification to the manuscript: e.g.

*This region was chosen for its combination of high inter-annual polynya activity and high spatio-temporal coverage with Sentinel-1 data. Results are expected to be transferable to other polynya regions in the Antarctic.*

**L54: At this stage, It strikes me that it might be better to describe input data before discussing the methods (i.e., move section 2.2 to here). This may have just been my personal preference though.**

We ill change the order.

**L112: This parapgraph needs more explanation. E.g., the 29-23. . . metric. Also, why are some numbers in bold type? What are the 35 epochs? 100 whats in a batch? Huber needs a capital H too.**

Due to overall changes in the manuscript related to comments by the Reviewer and a change in the processing software, this part will also change as described in a comment below. However, numbers remain also in the updated manuscript and refer to the number of hidden layers and associated neurons in the neural network architecture (here the autoencoder). For the above example, this refers to a total of seven layers with an input and output layer consisting of the 39 input variables; two hidden layers with 23 and 10 neurons respectively on each side of the dimensional reduced layer consisting of three neurons. The bold face numbers (on the left) highlight the encoder part of the autoencoder, which is used for the dimensional reduction. The decoder part is only used for the training of the autoencoder and not used afterwards. We added a clarification to the manuscript. For further reading we suggest the standard textbook by Goodfellow et al. (2016); https://www.deeplearningbook.org/.

**Section 2.2: You need to describe the version of the MODIS products – particularly the MOD29 product. This determines which MOD35 version went into the product. There have been some significant improvements to MOD35 over the years, so it's important to document which version. There is no description of the gridding or MODIS destriping/de-bowtie here. These must have huge influence on the performance of the algorithm, so a description of these processes is needed, in my opinion. Many of the channels used suffer strongly from detector striping in particular.**

We appreciate the reviewer for pointing these shortcomings out to us. The MOD/MYD29 version used is Version 6. We will also add this information to the manuscript.

The gridding is described in the manuscript (L145f), however, any additional pre-processing is not. The remainder of this comment is addressed in the process of following the suggestion by the reviewer concerning the cal/val.

**L140: What is the resolution of the IST product?**

Also 1km x 1km as the MOD021KM. We will add this information to the manuscript.

**Section 2.2.1: The destriping description may fit better here.**

Please refer to comment above.

**L158: What about the increased atmospheric path encountered for high incidence angles – is that more important than the geometry distortion?**

Please refer to our answer to your cal/val comment below.

**L164: Does MOD/MYD29 also apply atmospheric correction to more accurately determine IST?**

According to https://nsidc.org/data/mod29, the MOD/MYD29 product is derived from MOD/MYD021KM product, which is using the TOA radiances, so no atmospheric correction is applied.

**L167: I guess you're developing this algorithm for coastal, latent heat polynyas. It might be good to make this clear here. I doubt it would work for offshore/sensible heat ones (which is fine)!**

The success of the proposed cloud discrimination depends on the temperature regime and textural properties. To our knowledge, the thin-ice retrieval generally works for all thin ice areas. Problems could arise for rather fast changing shapes of offshore polynyas,as a set of input parameters depends on differences between swaths. However, the referenced line number solely relates on the selection of the SAR reference data.

**L178: This sentence "Generally, " is somewhat ambiguous.**

We will remove this sentence.

**Fig 3: One sub-figure would benefit from including a distance scale.**

We will add one.

**Fig 3: "Examplary" typo.**

We will fix that one. Thanks!

**L219: The cal/val split was done on a point-wise basis? This seems a bit strange. Isn't the point of the cal/val split to ensure independence between the calibration and validation datasets by withholding at a more basic level, e.g., scene level? What I'm trying to say is, two neighbouring pixels are unlikely to be completely independent. So if there's a 75% chance of each pixel getting into the training dataset, then it's pretty much guaranteed that the validation dataset won't be particularly independent of the training dataset. Could you comment on this?**

This is a very critical point and the reviewer is correct in his/her understanding and in bringing this up.

Due to this comment and the additional time we had due to the extensive discussion phase, we decided for a re-do of this part of the manuscript and the approach (as it also touches the time consuming part of manual categorization). Additionally, we also decided for a change towards a different machine-learning software environment inside the *R* environment. By implementing the *keras* package (a frontend for the well-known *tensorflow* backend), we also allow for a better distribution of the final neural network to potentially interested colleagues/collaborators as it can be easily transferred and adopted to/by other *R* as well as *python* users. We hope this rather big change is also in the reviewers and the editors interest and within the scope of the review process.

Here, we would like to briefly summarize the made changes to the manuscript work flow as bullet points. However, while the general procedure and overall results have not changed, we think these changes lead to an overall clearer description of our algorithm development as well as clarify/handle other brought-up comments by both reviewers.

- Instead of doing a point-wise randomized split of all our data, we settled for a swath-based split as suggested by the Reviewer. This ensures independence between both data sets. We now, in total, used **eight** combinations of Sentinel1-A/B/MODIS collocation data as **calibration** and **four** as **validation** (we took two examples that were previously shown in Figs. 4 and 5 and 'moved' them to calibration).

- We simplified and clarified the selection procedure of these swaths by selecting only the one swath closest in time to the Sentinel1 acquisition that **covers the study area by at least 90%** and features **incidence angles equal or below 35deg in 65%** of the study area. Both measures ensure high quality for manual categorization of the MODIS data.

- We generated in total 18 combinations per identified swath with surrounding swaths (90% coverage and at least 60% coverage with incidence angels of 50deg and below) to ensure high variability in the predictors used for the classification. To account for striping in the MODIS data and clarify and streamline the overall algorithm description, **we limit the data from MODIS to channels 20, 25, 31, and 33 in addition to IST data as well as the GLCM metric**s. This reduces the inut features to in total 34.

- With this we generally followed the same setup and procedure as before in the manuscript but used the *keras* R environment to train and use the Autoencoder as well as the initial and final Neural Network classifiers.

The resulting new classifier provides results as capable as before, however, it accuracy assessment is more realistic than before and likely less prone to overfitting.

**L278: Again, the Fraser et al., 2009 reference which shows this in a flaw lead would be good to reference here.**

We will add that one.

**Fig 4, 5: There is a mismatch between the actual extent of the Brunt Ice Shelf and the masked version, based on the Rtopo product. This is due to ice shelf advection in the time between the creation of both products. In this case, there are both areas of ice shelf outside the mask, and areas of water/sea ice within the mask. Other highresolution coastal datasets have mitigated this by including a manually-updated ice shelf extent product on a regular basis (e.g., Fraser et al., 2020, ESSD Discussions,**

**https://doi.org/10.5194/essd-2020-99), but this level of mitigation is probably unwarranted here. However, could you comment on the effect this might have on the training algorithm?**

The oftentimes very cold temperatures on the ice shelf would actually result in a similar difficulty to be reproduced by the neural network as we see with the wide temperature range for clouds. This is why we employed an at least rough estimate of the ice shelf to exclude these areas during the training process. While small scale effects exist, as pointed out by the reviewer, the effect on the training success appears to be negligible.

**Fig 6: This is a great way of showing the seasonality in bias. However I'm still hanging out for a good old-fashioned scatterplot comparing these two datasets. This would show highly complementary information to your time series.**

Agreed. We will add a scatterplot to the figure.

**L286: I think the "average" metric you use here may not be the best way to highlight how much better your algorithm performs! Have you considered also using RMS difference?**

**L337: Again, the suggested RMS statistic would better highlight your improvement.**

While we agree with the reviewer that RMS is a statistical value of interest, the daily coverage differences of the area due to different cloud screenings between the two methods would supposedly dominate the RMS in resulting polynya area. Therefore, we assume it is probably not usable as a quality measure of the method after all. However, we will give it a try and think of potential additional measures to use. Nonetheless, we think the result shown in the current manuscript – that also older studies without the benefit of a better cloud screening still hold valuable information at least on an annual basis – is of interest and value to the scientific community.

**L303: Unclear which product that this statement corresponds to.**

We agree and will change the sentence to clearly refer to the IST swath data.

**L309: "Good agreement" between what and what?**

We will clarify this by adding: e.g. *between the OSCD and the MOD/MYD29 product* to the end of the stated sentence.

---

## Author Comment (AC2) · 2 Dec 2020

We would like to thank Reviewer 2 for his comments and appreciate the time and effort he/she put into his/her review of our manuscript. In the following, we would like to go through all comments/suggestions and reply to them or answer them point-by-point. Reviewer comments are put in **bold font**, our replies are colored and changes inserted to the manuscript are put in *italics*.

**Major comments:**

**There are many coastal polynyas around Antarctica. A map of sea-ice production in coastal polynyas derived from the PM data (Nihahshi and Ohshima, 2015; Nihashi et al., 2018) shows that the ice production in a coastal polynya near the Brunt Ice Shelf region is small. This indicates that this small polynya's impact on climate change is not so significant as the other larger polynyas. Why did the authors focus on this minor polynya as a study area? In the title and throughout the manuscript, as stated in "Antarctic sea ice", it gives the impression that this manuscript's results have been achieved as if they represent the entire Antarctic Ocean. I wonder that a result of the cloud mask from a small polynya study can represent the entire Antarctic coastal polynyas or that the result of this manuscript can be applied to other large major coastal polynyas, such as the Ross Ice Shelf Polynya.**

This point was brought up by both reviewers. The region was chosen because of earlier experience with the region through the corresponding author (see Paul et al. 2015a,b) as well as several benefits for the setup of the algorithm we would like to explain in the following. While this polynya is not a major player in, e.g., deep-water formation (as correctly pointed out) it is one of the most active regions in the Weddell Sea and similar in ice production to the much larger Ronne Ice Shelf polynya (see Paul et al. 2015b). The high activity as well as the good spatio-temporal coverage through Sentinel-1 led to the decision to start-off with this region. However, the approach is assumed independent of the selected region as it mainly depends on the received satellite TIR signals, i.e., the temperature differences between surface types and clouds. Due to the investigation of a complete freezing period, this is assumed to be comparable to other polynya regions and should also be independent of polynya shape/size. A study applying the proposed procedure to all Antarctic polynyas is currently under preparation.

Regarding the manuscript title, we are sorry that Reviewer 2 was slightly disappointed by not (yet) seeing the algorithm applied to the complete Antarctic. However, we think the title focuses more on the methodology (which is applied in the Antarctic) rather than a particular region and would leave it for now as is.

We will add a clarification to the manuscript: e.g.,

*This region was chosen for its combination of high inter-annual polynya activity and high spatio-temporal coverage with Sentinel-1 data. Results are expected to be transferable to other polynya regions in the Antarctic.*

**Minor comments:**
**P. 2, L. 22: Please correct "polyanya" to "polynya".**
**P. 2, L. 23: "both in, the Arctic ...". It would be "both in the Arctic ...".**

We will change both. Thanks for pointing those out!

**P. 3, Fig. 2: There are no linkages between ted characters of A-H and the manuscript. I felt that elaborating by following these in the manuscript would be helpful for readers.**

This is a very helpful suggestion and we will change the manuscript accordingly by adding references to these sub-points in the Data&Methods section.

**P. 4, L. 67: "ii) large number". It would be "ii) a large number".**
**P. 5, L. 103: "hyberbolic tangent". It should be "hyperbolic tangent".**

We will also change those. Thanks for pointing those out!

**P. 5, L. 112: Please explain the number: "39-23-10-3-10-23-39" and the meaning of the number shown by the bold character of "39-23-10-3".**

Due to overall changes in the manuscript related to comments by Reviewer 1 and a change in the processing software, this part was also changed. However, numbers remain also in the updated manuscript and refer to the number of hidden layers and associated neurons in the neural network architecture (here the autoencoder). For the above example, this refers to a total of seven layers with an input and output layer consisting of the 39 input variables; two hidden layers with 23 and 10 neurons respectively on each side of the dimensional reduced layer consisting of three neurons. The bold face numbers (on the left) highlight the encoder part of the autoencoder, which is used for the dimensional reduction. The decoder part is only used for the training of the autoencoder and not used afterwards. We added a clarification to the manuscript. For further reading we suggest the standard textbook by Goodfellow et al. (2016); https://www.deeplearningbook.org/.

**P. 8, L. 178: Please correct from ", 2003).Generally" to ", 2003). Generally". Insert a space.**
**P.9, L. 184: "... data is of ...". It would be "...data are of ...".**

Another double correction. Thanks!

**P. 10, Fig. 3c: Correspondence between cloud, open water/thin ice, and sea ice and color is hard to identify. For example, my suggestion is that clusters 3, 7, 18, and 23 that correspond to the cloud area are shown by similar colors that can clearly distinguish from open water/thin ice and sea ice areas. Also, do each of the 3, 7, 18, and 23 clusters that correspond to clouds reflect the type of cloud? Further, what is the white area that does not belong to any cluster in this figure?**

The figure only provides an examples subset of all 35 clusters that we generated for each swath, therefore some area remains white in this example. The different clusters pointed out by reviewer not necessarily correspond to a certain cloud type, but rather represent a mix of temperature and texture. Also, due to the nature of the employed soft clustering (fuzzy c-means) each pixel belongs to all clusters but only with a certain probability. Therefore, it is likely that some of these cloud clusters probably show quite high or similar probabilities to neighboring cloud clusters as the number of 35 total clusters is for certain settings too much.

Our thought for the colors was to show as much clusters as possible with maximum contrast in colors to still be able to distinguish them. However, the suggestion to summarize clusters of the same type with similar colors also has something to it and we might change that in the final manuscript version.

**P.11, L. 208: "... a FCM probability ...". It would be "... an FCM probability ...".**

We will correct this one.

**P. 11, L. 212: The authors defined threshold values of temperature. How did you define these values? Is there any physical background?**

As stated in the manuscript, the separation was needed to aid the machine learning approach in understanding the image composition, as clouds especially experience a very wide range of temperatures (e.g., in contrast so sea-ice and open-water/thin-ice areas). However, there was no physical background in selecting these thresholds. These were arbitrarily chosen based on the overall temperature distribution in the training data in order to keep a majority in the intermediate class but cover for both extreme ends with sufficient training examples.

**P. 13, L. 267: "negliable". It should be "negligible".**

We will also correct this one.

**P. 14, Fig. 4a, e, and i: A polynya area surrounded by red line: the authors described that the area was "manually picked". How did you define the polynya area?**
**P. 15, Fig. 5a, e, and i: Same as the above.**

We based our decision on the textural differences seen in the SAR image that can be associated with different ice types or open water.

**P. 15, bottom: "and MOD/MYD29 estimated". It would be "and MOD/MYD29 was estimated".**
**P. 17, L. 307: "... West of ...". It would be "... west of ...".**

We will also correct those last typos. Thanks you again!

---

## Author Response (AR1)

The very helpful comments and suggestions by Reviewer 1 are very much appreciated and we would like to thank him/her for the time and effort he/she put into this review of our manuscript. In the following, we would like to go through all comments/suggestions and reply to them or answer them point-by-point. Reviewer comments are put in **bold font**, our replies are colored and changes inserted to the manuscript are put in *italics*.

**In this paper, the authors present a significant advance in the accurate detection of cloud in polar regions using advanced machine-learning techniques, and go on to illustrate the utility of this algorithm by retrieving much better estimates of thin ice thickness in the vicinity of the Brunt Ice Shelf, West Antarctica. The novel approach and subsequent advances in cloud detection are sorely needed for polar MODIS users, and as such I believe this paper is highly suitable for publication in TC, after minor modifications outlined below.**

We appreciate the comment and are happy to reply to the made comments.

**L22-29: It would be nice to point to other polar MODIS applications which would benefit from such a better cloud mask. Examples include composite image generation, landfast ice mapping, possibly sea ice motion retrieval using image cross-correlation.**

We followed this suggestion and added the following studies to the text:

*Additionally, other MODIS applications would potentially benefit from an improved wintertime cloud masking. These applications comprise composite generation (e.g., Fraser et al., 2010, 2020), merged optical and passive microwave sensor applications (e.g., Ludwig et al., 2019), basin-wide lead detection from thermal-infrared data (e.g., Reiser et al., 2020), as well as sea-ice motion tracking through image cross-correlation.*

Ludwig, V., G. Spreen, C. Haas, L. Istomina, F. Kauker, & D. Murashkin (2019). The 2018 North Greenland polynya observed by a newly introduced merged optical and passive microwave sea-ice concentration dataset. The Cryosphere, 13(7), 2051–2073. doi:10.5194/tc-13-2051-2019

Reiser, F.; Willmes, S.; Heinemann, G. A New Algorithm for Daily Sea Ice Lead Identification in the Arctic and Antarctic Winter from Thermal-Infrared Satellite Imagery. Remote Sens, 2020, 12, 1957. https://doi.org/10.3390/rs12121957

Aulicino, G.; Sansiviero, M.; Paul, S.; Cesarano, C.; Fusco, G.; Wadhams, P.; Budillon, G. A New Approach for Monitoring the Terra Nova Bay Polynya through MODIS Ice Surface Temperature Imagery and Its Validation during 2010 and 2011 Winter Seasons. Remote Sens. 2018, 10, 366.

Fraser, A. D., Massom, R. A., and Michael, K. J.: Generation of high-resolution East Antarctic landfast sea-ice maps from cloud-free MODIS satellite composite imagery, Remote Sensing of Environment, 114, 2888–2896, http://www.sciencedirect.com/science/article/pii/S0034425710002221, 2010.

**L29: This problem has also been reported in coastal leads, e.g. Fig. 6 from Fraser et al., 2009. DOI: 10.1109/TGRS.2009.2019726**

We added the suggested reference.

**L40: Why was this study region chosen? Is this applicable for both flaw leads and nonlinear coastal polynyas (e.g., Terranova Bay)?**

This point was brought up by both reviewers. The region was chosen because of earlier experience with the region through the corresponding author (see Paul et al. 2015a,b) as well as several benefits for the setup of the algorithm we would like to explain in the following. While this polynya is not a major player in, e.g., deep-water formation (as pointed out by Reviewer 2) it is one of the most active regions in the Weddell Sea and similar in ice production to the much larger Ronne Ice Shelf polynya (see Paul et al. 2015b). The high activity as well as the good spatio-temporal coverage through Sentinel-1 led to the decision to start-off with this region. However, the approach is assumed independent of the selected region as it mainly depends on the received satellite TIR signals, i.e., the temperature differences between surface types and clouds. Due to the investigation of a complete freezing period, this is assumed to be comparable to other polynya regions and should also be independent of polynya shape/size. A study applying the proposed procedure to all Antarctic polynyas is currently under preparation.

We clarified our reasoning in the manuscript as follows:

*This region was chosen for its combination of high inter-annual polynya activity and high spatio-temporal coverage with Sentinel-1 data. Results are expected to be transferable to other polynya regions in the Antarctic.*

**L54: At this stage, It strikes me that it might be better to describe input data before discussing the methods (i.e., move section 2.2 to here). This may have just been my personal preference though.**

We kept the order as is. We think to give a brief introduction to the methods feels more appropriate by following up with the data that these are applied to.

**L112: This parapgraph needs more explanation. E.g., the 29-23. . . metric. Also, why are some numbers in bold type? What are the 35 epochs? 100 whats in a batch? Huber needs a capital H too.**

Due to overall changes in the manuscript related to comments by both reviewers and a change in the processing software, this part changed as described in a comment below. However, we tried to clarify these things in a bit more detail in the respective training sections, while keeping the method introduction parts rather free of these technicalities specifically related to this study. For further reading we suggest the standard textbook by Goodfellow et al. (2016); https://www.deeplearningbook.org/.

**Section 2.2: You need to describe the version of the MODIS products – particularly the MOD29 product. This determines which MOD35 version went into the product. There have been some significant improvements to MOD35 over the years, so it's important to document which version. There is no description of the gridding or MODIS destriping/de-bowtie here. These must have huge influence on the performance of the algorithm, so a description of these processes is needed, in my opinion. Many of the channels used suffer strongly from detector striping in particular.**

We appreciate the reviewer for pointing these shortcomings out to us. The MOD/MYD29 version used is Version 6. We added this information to the manuscript.

The gridding is described in the manuscript (L131f), however, any additional pre-processing is not. The remainder of this comment is addressed in the process of following the suggestion by the reviewer concerning the cal/val.

**L140: What is the resolution of the IST product?**

Also 1km x 1km as the MOD021KM. We added this information to the manuscript.

**Section 2.2.1: The destriping description may fit better here.**

Please refer to comment above.

**L158: What about the increased atmospheric path encountered for high incidence angles – is that more important than the geometry distortion?**

Please refer to our answer to your cal/val comment below.

**L164: Does MOD/MYD29 also apply atmospheric correction to more accurately determine IST?**

According to https://nsidc.org/data/mod29, the MOD/MYD29 product is derived from MOD/MYD021KM product, which is using the TOA radiances, so no atmospheric correction is applied.

**L167: I guess you're developing this algorithm for coastal, latent heat polynyas. It might be good to make this clear here. I doubt it would work for offshore/sensible heat ones (which is fine)!**

The success of the proposed cloud discrimination depends on the temperature regime and textural properties. To our knowledge, the thin-ice retrieval generally works for all thin ice areas. Problems could arise for rather fast changing shapes of offshore polynyas,as a set of input parameters depends on differences between swaths. However, the referenced line number solely relates on the selection of the SAR reference data.

**L178: This sentence "Generally, " is somewhat ambiguous.**

We removed this sentence.

**Fig 3: One sub-figure would benefit from including a distance scale.**

We added one to all figures where applicable.

**Fig 3: "Examplary" typo.**

We fixed that one. Thanks!

**L219: The cal/val split was done on a point-wise basis? This seems a bit strange. Isn't the point of the cal/val split to ensure independence between the calibration and validation datasets by withholding at a more basic level, e.g., scene level? What I'm trying to say is, two neighbouring pixels are unlikely to be completely independent. So if there's a 75% chance of each pixel getting**

**into the training dataset, then it's pretty much guaranteed that the validation dataset won't be particularly independent of the training dataset. Could you comment on this?**

This is a very critical point and the reviewer is correct in his/her understanding and in bringing this up.

Due to this comment and the additional time we had due to the extensive discussion phase, we decided for a re-processing of this part of the manuscript and the approach (as it also touches the time consuming part of manual categorization). Additionally, we also decided for a change towards a different machine-learning software environment inside the *R* environment. By implementing the *keras* package (a frontend for the well-known *tensorflow* backend), we also allow for a better distribution of the final neural network to potentially interested colleagues/collaborators as it can be easily transferred and adopted to/by other *R* as well as *python* users. We hope this rather big change is also in the reviewers and the editors interest and within the scope of the review process.

Here, we would like to briefly summarize the made changes to the manuscript work flow as bullet points. However, while the general procedure has not changed, we think these changes lead to an overall clearer description of our algorithm development as well as clarify/handle other brought-up comments by both reviewers.

- Instead of doing a point-wise randomized split of all our data, we settled for a swath-based split as suggested by the Reviewer. This ensures independence between calibration and validation data sets. We now, in total, used **16** combinations of Sentinel1-A/B/MODIS collocation data as **calibration** and **six** as **validation**.

- We simplified and clarified the selection procedure of these swaths by selecting only the one swath closest in time to the Sentinel1 acquisition that **covers the study area by at least 90%** and features **incidence angles equal or below 35deg in 65%** of the study area. Both measures ensure high quality for manual categorization of the MODIS data.

- We generated in total 7 combinations per identified swath with surrounding swaths (90% coverage and at least 60% coverage with incidence angels of 50deg and below) to ensure high variability in the predictors used for the classification. To account for striping in the MODIS data and clarify and streamline the overall algorithm description, **we limit the data from MODIS to channels 20, 25, 31, and 33 in addition to IST data as well as the GLCM metrics.** This reduces the input features to in total 33.

- With this we generally followed the same setup and procedure as before in the manuscript but used the *keras* R environment to train and use the Autoencoder as well as the initial and final Neural Network classifiers.

The resulting new classifier provides results as capable as before, however, it accuracy assessment is more realistic than before and likely less prone to overfitting. Through the changed setup of calibration and validation files, however, the performance in general changed, leading to different final results. Generally, we find no a substantial decrease in annual average PA in OSCD compared to MOD/MYD29 with at the same time a still also substantial increase in sub-daily coverage. The changes in PA are summarized in this additional figure provided only in this response.

[Figure]

*Figure 1: PA daily time series (NEW minus OLD algorithm)*

Figure 1 shows similar to Figure 6 in the manuscript the daily difference PA but now between the old and the new algorithm (NEW minus OLD). Generally, the new algorithm leads to lower daily PA estimates. blue/red colored bars highlight certain days that we investigated further to see whether this is a good change in the algorithm performance.

[Figure]

*Figure 2: Example swaths to DOY96 (6 April 2017)*

Examples are shown in Figure 2 & Figure 3, whereas the former shows a positive one, and the latter a negative one. These image sets comprise the swath IST (a) and the OSCD results of the new (b) and old (c) algorithm as well as the resulting daily composite TIT which is converted into PA in the manuscript. While the positive example (new algorithm provides a larger estimate), classifies a lot of the warm sea ice as well as some clouds as "thin ice", the processing step of using the energy balance model to estimate TIT limits the impact of this presumably false classification. However, in the negative example (old algorithm provides the way larger estimate), we see a lot of warm clouds/sea ice to be classified as "thin ice" with a substantial impact on the resulting daily composite.

[Figure]

*Figure 3: Example swaths to DOY149 (29 May 2017)*

**L278: Again, the Fraser et al., 2009 reference which shows this in a flaw lead would be good to reference here.**

We added that one.

**Fig 4, 5: There is a mismatch between the actual extent of the Brunt Ice Shelf and the masked version, based on the Rtopo product. This is due to ice shelf advection in the time between the creation of both products. In this case, there are both areas of ice shelf outside the mask, and areas of water/sea ice within the mask. Other highresolution coastal datasets have mitigated this by including a manually-updated ice shelf extent product on a regular basis (e.g., Fraser et al., 2020, ESSD Discussions, https://doi.org/10.5194/essd-2020-99), but this level of mitigation is probably unwarranted here. However, could you comment on the effect this might have on the training algorithm?**

The oftentimes very cold temperatures on the ice shelf would actually result in a similar difficulty to be reproduced by the neural network as we see with the wide temperature range for clouds. This is why we employed an at least rough estimate of the ice shelf to exclude these areas during the training process. While small scale effects exist, as pointed out by the reviewer, the effect on the training success appears to be negligible.

**Fig 6: This is a great way of showing the seasonality in bias. However I'm still hanging out for a good old-fashioned scatterplot comparing these two datasets. This would show highly complementary information to your time series.**

Agreed. We added a small scatterplot to the figure.

**L286: I think the "average" metric you use here may not be the best way to highlight how much better your algorithm performs! Have you considered also using RMS difference?**
**L337: Again, the suggested RMS statistic would better highlight your improvement.**

While we agree with the reviewer that RMS is a statistical value of interest, the daily coverage differences of the area due to different cloud screenings between the two methods would supposedly dominate the RMS in resulting polynya area. Therefore, we assume it is probably not usable as a quality measure of the method after all. However, due to an error found during the

re-calculation/coding process, numbers changed slightly and better reflect the substantial improvements with our methods – already with the average metric.

**L303: Unclear which product that this statement corresponds to.**
**L309: "Good agreement" between what and what?**

Both parts changed in the final manuscript version.

We would like to thank Reviewer 2 for his comments and appreciate the time and effort he/she put into his/her review of our manuscript. In the following, we would like to go through all comments/suggestions and reply to them or answer them point-by-point. Reviewer comments are put in **bold font**, our replies are colored and changes inserted to the manuscript are put in *italics*.

**Major comments:**

**There are many coastal polynyas around Antarctica. A map of sea-ice production in coastal polynyas derived from the PM data (Nihahshi and Ohshima, 2015; Nihashi et al., 2018) shows that the ice production in a coastal polynya near the Brunt Ice Shelf region is small. This indicates that this small polynya's impact on climate change is not so significant as the other larger polynyas. Why did the authors focus on this minor polynya as a study area? In the title and throughout the manuscript, as stated in "Antarctic sea ice", it gives the impression that this manuscript's results have been achieved as if they represent the entire Antarctic Ocean. I wonder that a result of the cloud mask from a small polynya study can represent the entire Antarctic coastal polynyas or that the result of this manuscript can be applied to other large major coastal polynyas, such as the Ross Ice Shelf Polynya.**

This point was brought up by both reviewers. The region was chosen because of earlier experience with the region through the corresponding author (see Paul et al. 2015a,b) as well as several benefits for the setup of the algorithm we would like to explain in the following. While this polynya is not a major player in, e.g., deep-water formation (as correctly pointed out) it is one of the most active regions in the Weddell Sea and similar in ice production to the much larger Ronne Ice Shelf polynya (see Paul et al. 2015b). The high activity as well as the good spatio-temporal coverage through Sentinel-1 led to the decision to start-off with this region. However, the approach is assumed independent of the selected region as it mainly depends on the received satellite TIR signals, i.e., the temperature differences between surface types and clouds. Due to the investigation of a complete freezing period, this is assumed to be comparable to other polynya regions and should also be independent of polynya shape/size. A study applying the proposed procedure to all Antarctic polynyas is currently under preparation.

Regarding the manuscript title, we are sorry that Reviewer 2 was slightly disappointed by not (yet) seeing the algorithm applied to the complete Antarctic. However, we think the title focuses more on the methodology (which is applied in the Antarctic) rather than a particular region and would leave it for now as is.

We clarified our reasoning in the manuscript as follows:

*This region was chosen for its combination of high inter-annual polynya activity and high spatio-temporal coverage with Sentinel-1 data. Results are expected to be transferable to other polynya regions in the Antarctic.*

**Minor comments:**
**P. 2, L. 22: Please correct "polyanya" to "polynya".**
**P. 2, L. 23: "both in, the Arctic ...". It would be "both in the Arctic ...".**
**P. 4, L. 67: "ii) large number". It would be "ii) a large number".**
**P. 5, L. 103: "hyberbolic tangent". It should be "hyperbolic tangent".**
**P. 8, L. 178: Please correct from ", 2003).Generally" to ", 2003). Generally". Insert a space.**

**P.9, L. 184: "... data is of ...". It would be "...data are of ...".**
**P.11, L. 208: "... a FCM probability ...". It would be "... an FCM probability ...".**
**P. 13, L. 267: "negliable". It should be "negligible".**
**P. 15, bottom: "and MOD/MYD29 estimated". It would be "and MOD/MYD29 was estimated".**
**P. 17, L. 307: "... West of ...". It would be "... west of ...".**

We either corrected the typos or removed the mentioned occurrences following suggestions of Reviewer 1. Thanks for pointing those out!

**P. 3, Fig. 2: There are no linkages between ted characters of A-H and the manuscript. I felt that elaborating by following these in the manuscript would be helpful for readers.**

This is a very helpful suggestion and we changed the manuscript accordingly by adding references to these sub-points in the respective (sub)sections. However, through the changes to the manuscript also following advises by the other reviewer, we changed parts of the figure and the workflow.

**P. 5, L. 112: Please explain the number: "39-23-10-3-10-23-39" and the meaning of the number shown by the bold character of "39-23-10-3".**

Due to overall changes in the manuscript related to comments by Reviewer 1 and a change in the processing software, this part was also changed. However, we tried to clarify these things in a bit more detail in the respective training sections, while keeping the method introduction parts rather free of these technicalities specifically related to this study. For further reading we suggest the standard textbook by Goodfellow et al. (2016); https://www.deeplearningbook.org/.

**P. 10, Fig. 3c: Correspondence between cloud, open water/thin ice, and sea ice and color is hard to identify. For example, my suggestion is that clusters 3, 7, 18, and 23 that correspond to the cloud area are shown by similar colors that can clearly distinguish from open water/thin ice and sea ice areas. Also, do each of the 3, 7, 18, and 23 clusters that correspond to clouds reflect the type of cloud? Further, what is the white area that does not belong to any cluster in this figure?**

The figure only provides an examples subset of all 35 clusters that we generated for each swath, therefore some area remains white in this example. The different clusters pointed out by reviewer not necessarily correspond to a certain cloud type, but rather represent a mix of temperature and texture. Also, due to the nature of the employed soft clustering (fuzzy c-means) each pixel belongs to all clusters but only with a certain probability. Therefore, it is likely that some of these cloud clusters probably show quite high or similar probabilities to neighboring cloud clusters as the number of 35 total clusters is for certain settings too much.

Our thought for the colors was to show as much clusters as possible with maximum contrast in colors to still be able to distinguish them. However, we changed the last panel and added two more to the figure to better reflect the resulting initial training data state, the final training data stage as well as the final classification result and hope this is also in accordance with the Reviewers intention.

**P. 11, L. 212: The authors defined threshold values of temperature. How did you define these values? Is there any physical background?**

As stated in the manuscript, the separation was needed to aid the machine learning approach in understanding the image composition, as clouds especially experience a very wide range of

temperatures (e.g., in contrast so sea-ice and open-water/thin-ice areas). However, there was no physical background in selecting these thresholds. These were arbitrarily chosen based on the overall temperature distribution in the training data in order to keep a majority in the intermediate class but cover for both extreme ends with sufficient training examples.

**P. 14, Fig. 4a, e, and i: A polynya area surrounded by red line: the authors described that the area was "manually picked". How did you define the polynya area?**
**P. 15, Fig. 5a, e, and i: Same as the above.**

We based our decision on the textural differences seen in the SAR image that can be associated with different ice types or open water.